# Regulation of Epicardial Cell Fate during Cardiac Development and Disease: An Overview

**DOI:** 10.3390/ijms23063220

**Published:** 2022-03-16

**Authors:** Cristina Sanchez-Fernandez, Lara Rodriguez-Outeiriño, Lidia Matias-Valiente, Felicitas Ramirez de Acuña, Francisco Hernandez-Torres, Estefania Lozano-Velasco, Jorge N. Dominguez, Diego Franco, Amelia Eva Aranega

**Affiliations:** 1Cardiovascular Development Group, Department of Experimental Biology, Faculty of Experimental Sciences, University of Jaén, 23071 Jaén, Spain; csfernan@ujaen.es (C.S.-F.); lrodrigu@ujaen.es (L.R.-O.); lmmatias@ujaen.es (L.M.-V.); fracuna@ujaen.es (F.R.d.A.); evelasco@ujaen.es (E.L.-V.); jorgendm@ujaen.es (J.N.D.); dfranco@ujaen.es (D.F.); 2Medina Foundation, Technology Park of Health Sciences, 18016 Granada, Spain; fhtorres@ugr.es; 3Department of Biochemistry and Molecular Biology III and Immunology, Faculty of Medicine, University of Granada, 18016 Granada, Spain

**Keywords:** epicardium, cardiac development, cardiac damage

## Abstract

The epicardium is the outermost cell layer in the vertebrate heart that originates during development from mesothelial precursors located in the proepicardium and septum transversum. The epicardial layer plays a key role during cardiogenesis since a subset of epicardial-derived cells (EPDCs) undergo an epithelial–mesenchymal transition (EMT); migrate into the myocardium; and differentiate into distinct cell types, such as coronary vascular smooth muscle cells, cardiac fibroblasts, endothelial cells, and presumably a subpopulation of cardiomyocytes, thus contributing to complete heart formation. Furthermore, the epicardium is a source of paracrine factors that support cardiac growth at the last stages of cardiogenesis. Although several lineage trace studies have provided some evidence about epicardial cell fate determination, the molecular mechanisms underlying epicardial cell heterogeneity remain not fully understood. Interestingly, seminal works during the last decade have pointed out that the adult epicardium is reactivated after heart damage, re-expressing some embryonic genes and contributing to cardiac remodeling. Therefore, the epicardium has been proposed as a potential target in the treatment of cardiovascular disease. In this review, we summarize the previous knowledge regarding the regulation of epicardial cell contribution during development and the control of epicardial reactivation in cardiac repair after damage.

## 1. Introduction

Over the years, the epicardium, a thin epithelial layer that covers the surface of the vertebrate heart, has been described as a critical source of cell and signaling necessary for cardiac development. During this process, the epicardium constitutes a population of multipotent progenitors that, after undergoing an epithelial-to-mesenchymal transition (EMT), migrate into the myocardium as epicardium-derived cells (EPDCs) [1]. EPDCs give rise to various cardiac cell types, including coronary vascular smooth muscle cells, cardiac fibroblasts, endothelial cells, and presumably a subpopulation of cardiomyocytes, thus contributing to complete heart formation [2,3,4,5]. However, it remains unclear as to how different cell types emerge from EPDCs and the molecular mechanisms underlying epicardial-derived cell fate during heart development. In addition, it is interesting to highlight that, although the epicardium remains as a quiescent cell monolayer after birth, epicardial cells in the adult heart can be reactivated in response to injury by upregulating the epicardial development genetic program [6,7]. Thus, following cardiac injury, the epicardium regulates cardiac wound healing and may have a potential role in tissue remodeling after heart damage. Therefore, the knowledge of the signals that mediate epicardial cell fate during development can help us to better understand the molecular mechanisms underlying the reactivation of epicardial cells after heart damage and would open novel potential therapeutic approaches in cardiovascular medicine.

In this review, we summarize how the epicardium is formed during development and examine the molecular signals that sustain the emergence of different EPDCs cell fates and the implications of epicardial-derived cell lineages in the response to cardiac injury and heart repair. Moreover, we summarize therapeutic strategies to promote heart regeneration and discuss why the epicardial cell layer can potentially be used in therapeutic approaches after cardiac damage.

## 2. Origin and Developmental Significance of the Epicardium

Commonly described as the outermost cell layer of the vertebrate heart, the epicardium is a conserved mesothelium which serves as a multipotent progenitor source during cardiac embryonic formation [8]. In the developing embryo, epicardium mainly originates from proepicardial cells, a cluster of cells that protrudes from the septum transversum, a folding of the lateral plate mesoderm [9,10,11]. While in chick and zebrafish embryos, cells from the proepicardium migrate toward the heart via extracellular matrix bridges, in mammals, the proepicardial cells migrate through the formation of free-floating cell aggregates or by direct contact with the myocardial surface (Figure 1) [12,13,14,15]. In most species, the epicardium is constituted by a single cell layer; however, in the human heart, the epicardium is a multilayer of mesothelial cells overlying connective tissue [16]. Therefore, between epicardial cell layers and the human myocardium there is a sub-epicardial space with elastic fibers and blood vessels that can accumulate adipose tissue deposits during the adult life [17]. Once the epicardium is fully formed, surrounding the myocardium almost completely, epicardial cells start to express a complex network of transcription factors, including the transcription factor 21 (Tcf21). It has been shown that in the absence of Tcf21, epicardial cells are retained in proepicardial precursor state and fail to form a cohesive polarized sheet, suggesting that this transcription factor is necessary for normal epicardial development [18,19,20,21]. Furthermore, other transcription factors are involved in the epicardium formation, such as the zinc finger transcription factor Wilms’ tumor 1 (WT1), required to maintain epicardial adhesion and integrity by the transcriptional activation of alfa4integrin (Itga4), or T-box transcription factor 18 (Tbx18), which maintains the epicardial progenitor status [22,23]. It has been described, in murine epicardial cells, that both epicardial transcription factors, Wt1 and Tbx18, can bind the *Slug* promoter region and regulate its expression, controlling the epicardial–mesenchymal balance. Thus, while Wt1 maintains epicardial properties and inhibits epicardial EMT through the downregulation of Slug expression, Tbx18 has the opposite effect, upregulating Slug expression and the epicardial epithelial-to-mesenchymal transition [24].

Far from being only one epithelial layer wrapping the heart, the epicardium is a crucial source of cells, ECM components, and paracrine factors required for the formation of compact myocardium and coronary vasculature patterning [25].

## 3. Regulation of Epicardial Cell-Derived Contribution during Heart Development

As development progresses, a subset of epicardial cells undergoes epithelial-to-mesenchymal transition (EMT), delaminating and migrating into the subepicardial space and invading the myocardium (Figure 2A). Those populations of epithelial cells, called epicardium-derived progenitor cells (EPDCs), lose their cell–cell adhesions and their apical-basal polarity acquiring migratory and invasive characteristics of mesenchymal stem cells [26,27,28]. To date, it is still not clear whether some epicardial cells are specified to acquire competence for EMT or if all epicardial cells have the potential to undergo EMT. It has been shown that epicardial cell division and spindle orientation seem to determine whether epicardial cell remain in the epicardium layer or undergoes EMT and migrates towards the myocardium. In this context, β-catenin seems to have an important role as it is required to establish epicardial cell polarity and its mutation randomized spindle orientation and reduced epicardial EMT [29].

The epithelial-to-mesenchymal transition (EMT) is a complex event highly regulated by different transcription factors, such as Snai1, Snai2, Twist1, or Hand2, that mediate the change from epithelial cells into a mesenchymal phenotype and are implicated in mesenchymal cell proliferation and migration [26,27,28,30,31,32]. In the same way, some TGFβ and FGF superfamily members promote epicardial EMT and EPDC motility [33,34]. In this regulated process, genes encoding epithelial adhesion molecules such as E-cadherin must be repressed while other genes necessary for extracellular matrix production and migration such as N-cadherin or fibronectin must be activated [26]. Extracellular matrix remodeling also plays an important role during EPDC invasion of the myocardium; for instance, the nuclear factor of activated T cell family of transcription factor, Nfact1, is required for cathepsinK (Ctsk) expression, a potent collagenase that facilitates cell migration [35].

Once in the myocardium, the delaminated EPDCs can be differentiated into specialized cells such as vascular smooth muscle cells or cardiac fibroblasts, which contribute to myocardial integrity by constructing the coronary vasculature [2,3]. The endothelial cells (ECs), whose developmental origin remains controversial, are a major component of coronary vessels. Endothelial cells of the coronary vasculature have been previously defined as non-epicardial-derived, and many studies have suggested that, in mammals, coronary ECs originate from the sinus venosus and ventricular endocardium [22,36,37,38]. However, in recent years, an extracardiac cell contribution to coronary vascular morphogenesis has been reported in mice. Hence, a subpopulation of endothelial cells in the coronary vasculature has been described as a population of endothelial progenitors arising from septum transversum/proepicardium mesothelial cells [9]. In the same fashion, during chick embryo development, Wt1-derived epicardial cells have been shown to give rise to coronary endothelial cells [22,36,39].

EPDC-derived lineage determination is a precisely regulated process mediated by complex interactions among signaling pathways and transcription factors, which can act as transcriptional activators and/or repressors [40]. For instance, *Pdgfra* expression promoted by Hand2 is required for epicardium-derived cardiac fibroblast specification, which is one of the most numerous cell populations in the heart, while *Pdgfrb* activation induces EPDCs differentiation into vascular smooth muscle cells [32,41,42,43,44,45]. Tcf21 regulates EPDC differentiation into smooth muscle and fibroblast lineages. However, the joint action of this transcription factor and RA signaling in pre-migratory EPDCs has a repressive role in the differentiation of the EPDC-derived vascular lineage, which finally adopt a fibrotic phenotype [19,36]. Furthermore, in the epicardium, Notch signaling directs EPDC differentiation into smooth muscle cells, whereas the transcription factor Tbx18 maintains epicardial cell identity by acting as a transcriptional repressor of this cell fate [2,46].

Despite of the fact that multiple cell types arise from epicardial progenitors, the timing and regulation in the emergence of its cellular derivatives as well as cell lineage determination are not fully characterized. Moreover, although some studies suggest that EPDC lineages are specified prior to epicardial EMT and arise from different populations; whether the diverse epicardial-derived cells lineages come from a common or distinct epicardial progenitor populations remains unsolved [47,48]. It has been proposed that the heterogeneity attributed to the origin and cellular composition of the proepicardium influences the multiple cell fates of EPDCs during heart development; however, a deeper understanding is needed to unlock the full potential of the epicardium as a source of different cell types that complete cardiac development [49]. Generation of reporter mouse lines and genetic lineage-tracing models have facilitated the identification of different cell lineages arising from the epicardium; for example, the majority of Wt1Cre- and Tcf21-based lineage-positive epicardial cell derivatives give rise to fibroblasts and vascular smooth muscle cells [36,50,51,52,53]. Although the contribution of EPDCs to endothelial cell and cardiomyocyte lineages remains under debate, recent reports have revealed that a minor fraction of coronary endothelium cells are derived from the epicardial GATA5 and myocardial cTnT lineages [4]. Furthermore, a small population of embryonic cardiomyocytes that expresses WT1 at low levels has recently been described; these WT1 lineage-derived cardiomyocytes are similar to a progenitor population named juxta-cardiac field (JCF), identified by single-cell RNA sequencing of mouse embryonic hearts [5,54]. Nevertheless, many discrepancies observed in the differentiation potential of EPDCs may be due to the fact that most of the promoters used in epicardial lineage-tracing models are not uniformly expressed and have a restrictive temporal expression pattern in the epicardium [5,55]. In this regard, a comparative study performed by Carmona et al. in four different epicardial lineage-tracing systems (WT1^Cre^, GATA5^Cre^, G2-GATA4 enhancer, and cTnT^Cre^) demonstrated that, although WT1^Cre^ driver seems useful in tracing the cell fate of EPDCs before E13.5, the interpretation of some WT1-lineage traced analyses can be difficult at later embryonic stages considering that later de novo expression may increase the proportion of Wt1-lineage cells [4].

In addition to its direct cellular contribution to cardiac development, the epicardium is a source of paracrine signals that serves as an essential secretory hub to maintain vascular and inflammatory cell stability during heart formation [56]. For instance, retinoic acid-mediated signaling induces the expression of FGF-9 and FGF-2 in epicardial cells, two mitogenic signals that facilitate myocardial growth [57,58,59]. In the same way, loss of FGFR1 in quail embryos reduces the myocardial invasion of epicardial cells [60]. The epicardial layer and EPDCs can also produce chemokines, such as C-X-C motif chemokine 12 (CXCL12), that regulate coronary vessel maturation and patterning [61,62]. TGFβ1 and TGFβ2 epicardial autocrine secretion stimulates the loss of epithelial phenotype and smooth muscle cell differentiation; those TGFβ-mediated effects are dependent on ALK5 activity and require p160 rho kinase [63]. When heart formation is complete, the epicardium downregulates many of its genetic markers, stops proliferating, and forms a continuous layer of cells with squamous morphology [64,65].

## 4. Epicardial Response to Cardiac Damage

The postnatal epicardium maintains a quiescent state; however, in response to cardiac damage, this epithelial layer suffers a reactivation initiating an embryonic-like response, including upregulation of Wt1 and EMT marker genes such as Tbx18 or Snai1 [66,67] (Figure 2B). Subsequently, epicardial cells undergo EMT and migrate into the underlying tissue contributing to post-injury repair; both actions appear to be crucial to facilitate heart regeneration [8,68]. Although the regeneration capacity of the mammalian heart is very limited, some studies have shown that the neonatal mouse heart can fully regenerate after injury [69,70,71,72]. This ability, restricted to the first week of mice life, revealed that, within a short time window, the heart retains a robust regeneration capacity; nevertheless, it is still uncertain as to whether reactivated postnatal epicardial cells sustain their potency and provide precursors of different cardiac lineages [22,47,71,73]. By analyzing the regeneration potential in neonatal mouse, Cai et al. observed that, after an apical heart resection, cardiac tissue can regrow and present smooth muscle cells and cardiomyocytes derived from Tbx18^+^ epicardial cells within the injured area [74]. However, the low number of Tbx18^+^-derived cells generated after injury were not sufficient to sustain successful cardiac regeneration, suggesting a decreased regeneration potential after birth [74]. According to this idea, it is well described that the adult mammalian heart exhibits an insufficient regenerative capacity due to the relative inability of mature cardiomyocytes to reenter the cell cycle and proliferate to repair ischemic tissue [75,76]. However, after cardiac injuries such as myocardial infarction or ischemic events, the expression of some developmental genes, such as *Wt1*, *Tbx18*, *Raldh1*, *Snai1*, or *alfaSMA*, are upregulated throughout the entire adult epicardium [67]. In a mouse myocardial infarction model, the spatio-temporal changes in genes expression have been analyzed, describing a peak 3 days after damage and subsiding after 2 weeks [66]. This epicardial reactivation seems to be required for both fetal and adult cardiac repair; however, the different molecular mechanisms underlying this epicardium-wide response are not well characterized [77]. In this regard, some studies have suggested that hypoxia and HIF signaling may work as regulators that trigger epicardial cell reactivation after cardiac damage [78,79].

Aside from the activation of genetic embryonic programs, reactivated epicardial cells become proliferative and form an expanded layer of EPDCs that constitutes a subepicardial niche that amplifies epicardial paracrine signals that promote the growth and survival of coronary vessels [67]. Furthermore, this multi-cell layer favors an adaptive immune regulation that eliminates dead cells and extracellular fragments from the injured area, allowing cell repopulation of the damage zone and promoting the cardioprotective and regenerative responses to cardiac injury [80,81]. Interestingly, some studies showed that the reactivated adult epicardium may also be able to preserve similar cellular plasticity to that observed during heart development. However, the use of lineage tracing models has revealed that the majority of newly differentiated cells that appear following cardiac injury arise from preexisting cardiac fibroblasts, endothelial cell populations, and/or smooth muscle cell populations [74,82,83,84,85,86,87]. Otherwise, in the mice model of atrial cardiomyopathy WT1^CreERT2/+^; ROSA26-tdT^+/−^, epicardial progenitor cells reactivate to differentiate into myofibroblasts during tissue remodeling [88]. This cell population, which progressively produces collagen, tries to stabilize the myocardial wall replacing contractile tissue with a fibrotic scar that is crucial for preventing further damage and the fatal rupture of the ventricular wall [89,90].

In animal models that display potential for cardiac regeneration, the fate and function of EPDCs during development and cardiac repair are modulated by extracellular matrix composition [91]. For instance, cardiac resection injury in newt heart induces epicardial expression of tenascin-C and hyaluronic acid, two matrix components that adapt the tissue stiffness to facilitate the EPDC migration throughout the myocardium [92,93]. Moreover, Missinato et al. have described the role of hyaluronic acid and its receptor, hyaluronan-mediated motility receptor (Hmmr), in regulating epicardial cell proliferation and migration through focal adhesion kinases (FAK) and Src kinases in the regenerating zebrafish heart [94].

## 5. Control of Epicardial Reactivation and Cardiac Repair: A Double-Side Tool

Although in the last decade there has been growing knowledge about the contribution of the epicardium during development and after cardiac injury, the way in which this cardiac cell layer contributes to disease progression or repair in the adult heart remains unclear. For instance, genetic depletion of epicardial cells in adult zebrafish inhibited cardiomyocyte proliferation and decreased muscle regeneration after a partial ventricular resection [95]. In this context, R-Spondin 1 (RSPO1) is a factor with potential to influence the regenerative response after injury; known to activate cell proliferation and to promote compact myocardium development, this protein is restricted to epicardial cells and seems to regulate angiogenesis during neonatal and heart regeneration in zebrafish [96,97,98,99]. However, Huang et al. showed that inhibition of CCAAT/enhancer binding protein (C/EBP) transcription factors in the adult murine epicardium reduced scar formation and neutrophil infiltration after injury: two key processes in undergoing appropriate cardiac remodeling after damage [100].

On the other hand, it is known that the adult epicardium reactivation following cardiac injury is also accompanied by the secretion of paracrine factors that can contribute to the repair response [67]. Thus, the Cxcl12b-Cxcr4a signaling in the epicardium of adult zebrafish was found to promote revascularization of the damaged area after a cardiac cryoinjury [101]. In several mouse models, such as Wt1^CreERT2/+^; Rosa26 ^mTmG/+^ mice, FGF2 and VEGFA secreted by EPDCs increased vessel density after myocardial infarction, suggesting that paracrine factors secreted by the reactivated epicardium can be a useful tool to increase the cardiac vasculature after damage [67]. Follistatin-like 1 (FSTL-1), a factor present within the secretome of epicardial cells, has been described as a potent cardiogenic factor due to its ability to induce cardiomyocyte proliferation when locally applied onto the infarcted area of mice hearts [102].

Additionally to pro-regenerative effects, recent reports have described a role for reactivated epicardium as a partial coordinator of cellular and paracrine inflammatory responses allowing regulatory T cells recruitment via epicardial Hippo signaling [81,103]. The massive loss of millions of cardiomyocytes after cardiac injury causes the release of damage-associated molecular pattern molecules (DAMPs) that act as endogenous signals alerting the immune system. Thus, after tissue injury, epicardial activation modulates an adaptive immune response triggering inflammatory cell infiltration to clear dead or dying cells. Although this is a compensatory mechanism that contributes to heart remodeling, excessive inflammatory reaction and chronic inflammation can enhance activation of proapoptotic signaling pathways, extensive formation of fibrosis, and cardiac dysfunction. It is important to highlight that the human heart is surrounded by an epicardial adipose tissue that originates from the epicardium in a process mediated by the metabolic regulator peroxisome proliferator-activated receptor gamma (PPAR- γ) [104]. In this context, the epicardium acts as a central regulator of the balance between fat and fibrosis accumulation; for instance, Activin-A, secreted by epicardial adipose tissue, induces a fibrotic phenotype of the atrial and ventricular myocardium that can lead to electrical dissociation and focal fibrillation waves [105,106,107,108]. In fact, the generation of an excessive epicardial-derived adipose tissue around the atria has prognostic value and correlates with the evolution of some pathologies such as coronary artery disease or arrhythmogenic right ventricular cardiomyopathy [109,110]. For instance, during atrial cardiomyopathy, the epicardial adipose tissue reactivation and expansion generates subsets of EPDCs that can maintain their adipogenic potential or differentiate, via Angiotensin II, into myofibroblasts [88].

## 6. The Epicardium and Future Perspectives for Cardiac Repair

Cardiovascular disease remains the main cause of death worldwide and, despite advances in cardiology having led to a large number of patients surviving to heart stroke, coronary heart disease, or heart failure, the irreversible cardiac muscle damage results in a growing population living with chronic heart diseases [111]. During embryonic development, the epicardium plays a significant role in the formation of the heart, contributing to the formation of different cardiac cell types, playing a role in the remodeling of the subepicardial matrix, and acting as an important source of paracrine stimulation [8,20,56,71]. Although many of the embryonic epicardial capabilities are conserved in the adult injured heart, the contribution of the epicardium to cardiac remodeling after damage appears to occur less efficiently compared to its role during development. To date, it is still not clear as to whether specific epicardial cell subpopulations can participate in cardiac remodeling or whether distinct cell types residing within the epicardial layer have determined capabilities during the reparative response [112]. Furthermore, in terms of transcriptional regulatory networks and lineage determination mechanisms, much is yet to be learned, although a recent single-cell RNA sequencing study in adult mouse hearts after myocardial infarction identified transcriptionally independent epicardial cell populations with different roles in secretion of paracrine factors, modulation of the innate immune response, or tissue regeneration [48].

Different approaches have been proposed to address cardiac pathology, from induction of cardiomyocyte proliferation or reprogramming of the fibroblast population to inhibition of the pro-inflammatory response that accompanies cardiac scar formation or damaged area revascularization [113,114,115]. These therapeutic strategies are just a sample of the ongoing research against cardiovascular disease. Nevertheless, it should not be forgotten that a deep understanding of how epicardial cell populations and/or subpopulations are determined during development as well as the knowledge of the molecular mechanisms underlying epicardial cell fate decisions in the embryo may help us to decipher the mechanisms by which epicardial cell populations participate in cardiac repair processes in the adult heart.

## Figures and Tables

**Figure 1 ijms-23-03220-f001:**
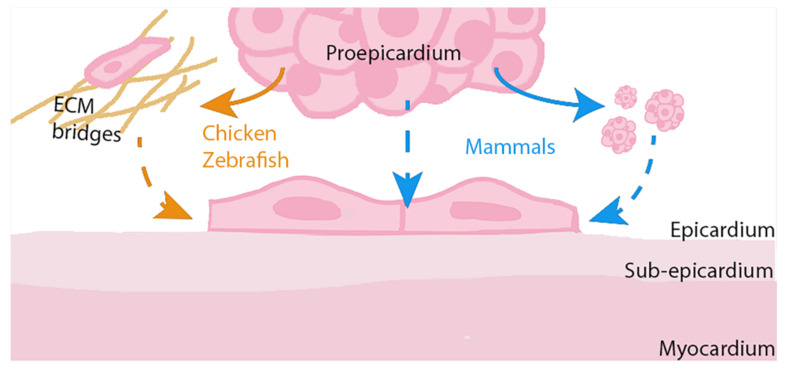
Epicardial development across species. In chick and zebrafish embryos, proepicardial cells migrate to the myocardial surface by formation of extracellular matrix (ECM) bridges (orange arrows). In mammals, the proepicardial cells migrate through the formation of free-floating cell aggregates or by direct contact with the myocardial surface (blue arrows).

**Figure 2 ijms-23-03220-f002:**
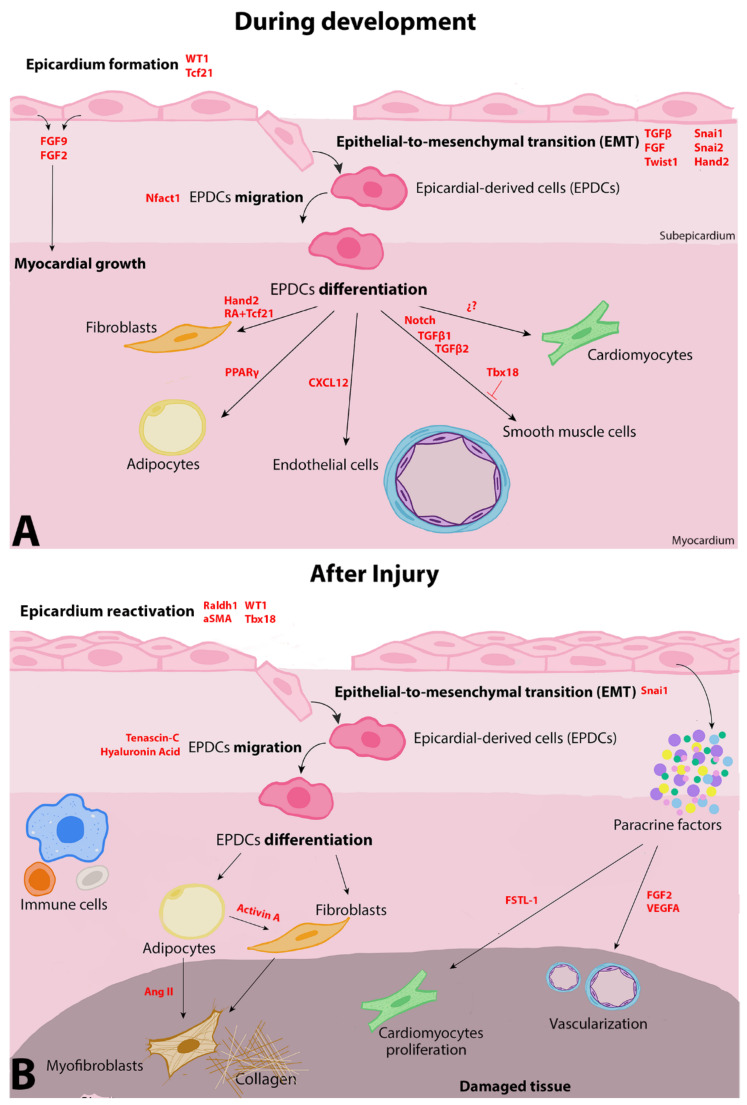
Cellular and molecular interactions during epicardial development and after injury. (**A**) A subset of epicardial cells, called epicardium-derived progenitor cells (EPDCs), undergo epithelial-to-mesenchymal transition (EMT); delaminate and migrate into the subepicardium; and differentiate into different cell types: fibroblasts, adipocytes, endothelial cells, vascular smooth muscle cells, and potentially cardiomyocytes. Many factors (red) have been described to be involved in driving epicardial EMT and EPDC migration and differentiation. (**B**) After cardiac injury, reactivated epicardial cells express genetic embryonic programs, undergo EMT, and migrate into the underlying tissue, where they differentiate into adipocytes or fibroblasts. Furthermore, the epicardium reactivation is accompanied by the secretion of paracrine factors and by an adaptive immune response.

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
