# Peer review of "Regulation of Epicardial Cell Fate during Cardiac Development and Disease: An Overview"

_ijms, 2022, doi:10.3390/ijms23063220_

Round 1
Reviewer 1 Report
Helpful and interesting review.
Very good work.
I have included some suggestions to improve the fluency and the English language.
Abstract
Line 19: remove comma and say “Although several lineage trace studies......”
Line 21: replace with “....the molecular mechanisms underlying epicardial cell heterogeneity remain not fully understood.”
Introduction
Line 37: it would be better to rearrange as “However, it remains unclear how different cell types emerge from EPDCs and the molecular mechanisms underlying epicardial-derived cell fate during heart development.”
Line 47: replace with “....potential therapeutic approaches in cardiovascular medicine.”
Line 65: replace with “....there is a sub-epicardial space...”
Line 87: probably it is better to say “..., delaminating and migrating into...”
Line 112-114: rearrange as follows “The endothelial cells (ECs), whose developmental origin remains controversial, are a major component of coronary vessels.”
Line 114: say “Endothelial cells of the coronary vasculature...”
Line 118: replace with “...has been reported in mice.”
Line 124: add comma “...and transcription factors, which can act...”
Line 128-129: rearrange as “Tcf21 regulates EPDC differentiation into smooth muscle and fibroblast lineages. However, the joint action...”
Line 139-140: replace with “...from different populations, whether the diverse epicardial-derived cells...”
Line 144: say “...to unlock the full potential of the epicardium as a source of...”
Line 156: replace with “...may be due to the fact that...”
Line 160: replace with “...demonstrated that, although WT1Cre driver seems useful in tracing the cell fate of EPDCs before E13.5, the interpretation of some WT1-lineage traced analysis can be difficult at later embryonic stages considering that later de novo expression may...”
Line 207: replace with “...and form an expanded layer...”
Line 215: replace with “...of newly differentiated cells...”
Line 227: rearrange as follows “Besides, Missinato et al. have described the role of hyaluronic acid and its receptor-hyaluronan-mediated motility receptor (Hmmr) in regulating epicardial cell proliferation and migration through focal adhesion kinases (FAK) and Src kinases in the regenerating zebrafish heart [94].”
Line 234: replace with: “Although in the last decade there has been...”
Line 236-238: replace with “For instance, genetic depletion of epicardial cells in adult zebrafish inhibited cardiomyocyte proliferation and decreased muscle regeneration after partial ventricular resection [95].”
Line 245: replace with “...after injury: two key processes to undergo appropriate cardiac remodelling....”
Line 248: replace with “On the other hand, it is known that adult epicardium reactivation following cardiac injury is also accompanied by...”
Line 281: replace with “..., via Angiotensin II,...”
Line 285: replace with “Cardiovascular disease remains...”
Line 286-287: replace with “...advances in cardiology have led to a large number of patients surviving these pathologies, the irreversible cardiac muscle damage results in...”
Line 306: replace with: “These therapeutic strategies are just a sample of the ongoing research against cardiovascular disease.”
Line 310: remove comma “...in the embryo may help us decipher...”
Author Response
REPLAY TO REVIEWER 1
We would like to thank the reviewer for careful and thorough reading of this manuscript and for the thoughtful comments and constructive suggestions, which help to improve the quality of this manuscript. Following your suggestion, we have corrected point by point the manuscript accordingly. Please, note that all corrections have been highlighted in bold in the new version of the manuscript.
Abstract
-Line 19: In accordance with reviewer comments in the new version of the manuscript comma has been removed and now says “Although several lineage trace studies......”
-Line 21: In accordance with reviewer suggestion in the new version of the manuscript line 21 has been replaced with “....the molecular mechanisms underlying epicardial cell heterogeneity remain not fully understood.”
Introduction
-Line 37 has been rearranged in accordance with reviewer suggestion as follow “However, it remains unclear how different cell types emerge from EPDCs and the molecular mechanisms underlying epicardial-derived cell fate during heart development.”
-In the new version of manuscript Line 47 has been replaced with “....potential therapeutic approaches in cardiovascular medicine.”
-In the new version of manuscript Line 65 has been replaced with “....there is a sub-epicardial space...”
-In the new version of manuscript Line 87 now says “..., delaminating and migrating into...”
-In the new version of manuscript Line 112-114 has been rearranged as follow “The endothelial cells (ECs), whose developmental origin remains controversial, are a major component of coronary vessels.”
-In the new version of manuscript Line 114 now says “Endothelial cells of the coronary vasculature...”
-In the new version of manuscript Line 118 has been replaced with “...has been reported in mice.”
-In the new version of manuscript Line 124 comma has been added as follow “...and transcription factors, which can act...”
-In the new version of manuscript Line 128-129 has been rearranged as “Tcf21 regulates EPDC differentiation into smooth muscle and fibroblast lineages. However, the joint action...”
-In the new version of manuscript Line 139-140 has been replaced with “...from different populations, whether the diverse epicardial-derived cells...”
-In the new version of manuscript Line 144 now says “...to unlock the full potential of the epicardium as a source of...”
-In the new version of manuscript Line 156 has been replaced with “...may be due to the fact that...”
-In the new version of manuscript Line 160 has been replaced with “...demonstrated that, although WT1Cre driver seems useful in tracing the cell fate of EPDCs before E13.5, the interpretation of some WT1-lineage traced analysis can be difficult at later embryonic stages considering that later de novo expression may...”
-In the new version of manuscript Line 207 has been replaced with “...and form an expanded layer...”
-In the new version of manuscript Line 215 has been replaced with “...of newly differentiated cells...”
-In the new version of manuscript Line 227 has been rearranged as follow “Besides, Missinato et al. have described the role of hyaluronic acid and its receptor-hyaluronan-mediated motility receptor (Hmmr) in regulating epicardial cell proliferation and migration through focal adhesion kinases (FAK) and Src kinases in the regenerating zebrafish heart [94].”
-In the new version of manuscript Line 234 has been replace with: “Although in the last decade there has been...”
-In the new version of manuscript Line 236-238 has been replace with “For instance, genetic depletion of epicardial cells in adult zebrafish inhibited cardiomyocyte proliferation and decreased muscle regeneration after partial ventricular resection [95].”
-In the new version of manuscript Line 245 has been replaced with “...after injury: two key processes to undergo appropriate cardiac remodelling....”
-In the new version of manuscript Line 248 has been replaced with “On the other hand, it is known that adult epicardium reactivation following cardiac injury is also accompanied by...”
-In the new version of manuscript Line 281 has been replaced with “..., via Angiotensin II,...”
-In the new version of manuscript Line 285 has been replaced with “Cardiovascular disease remains...”
-In the new version of manuscript Line 286-287 has been replaced with “...advances in cardiology have led to a large number of patients surviving these pathologies, the irreversible cardiac muscle damage results in...”
-In the new version of manuscript Line 306 has been replaced with: “These therapeutic strategies are just a sample of the ongoing research against cardiovascular disease.”
-In the new version of manuscript Line 310 comma has been removed and now reads “...in the embryo may help us decipher...”

Reviewer 2 Report
In this overview on the topic of epicardial cell fate in development and disease by Sanchez-Fernandez et. al. the authors provide a comprehensive description on the major studies that in this area. The review is well written and covers topics associated with epicardial cell function in heart development and in regeneration/repair. The authors further explore in great detail the studies that uncover how some epicardial cells can support the heart through differentiation into distinct cell types such as fibroblasts, smooth muscle cells and endothelial cells. Overall, this review is of great interest to researchers interested in learning about the epicardium and its functions.
One minor suggestion to improve on this excellent review is to include a diagram of the developmental source of epicardial cells. The authors discuss on where epicardial cells are derived from in development and describe differences in various species. This can be illustrated.
Other minor comments include some grammatical errors:
1) Line 202 states that “adult cardiac regeneration” exists, but maybe “repair” is more accurate if they are referring to mammals.
2) Line 116: No S needed at the end of originates
3) Line 286: this sentence is somewhat unclear as to what they mean by “…despite advances in cardiology have led a large number of patients surviving to these pathologies,…” Can the authors go into details about these pathologies
4) Line 281: Angiotensin instead of “Angiotensina”
Author Response
REPLAY TO REVIEWER 2
We would like to thank the reviewer for careful and thorough reading of this manuscript and for the thoughtful comments and constructive suggestions, which help to improve the quality of this manuscript. Following your suggestion, we have corrected point by point the manuscript accordingly. Please, note that all corrections have been highlighted in bold in the new version of the manuscript.
One minor suggestion to improve on this excellent review is to include a diagram of the developmental source of epicardial cells. The authors discuss on where epicardial cells are derived from in development and describe differences in various species. This can be illustrated.
-In accordance with reviewer suggestion a new figure including a diagram of the developmental source of epicardial cells has been added in the new version of the manuscript.
1) The Line 202 in the new version of the manuscript now states that “adult cardiac repair” exists.
2) In agreement with reviewer suggestion, in the Line 116 of the new version of the manuscript S has been removed at the end of originates
3) In accordance with reviewer suggestion, in Line 286 of the new version of the manuscript now reads : “ this sentence is somewhat unclear as to what they mean by “…despite advances in cardiology have led a large number of patients surviving to these pathologies,…” Can the authors go into details about these pathologies
4) Line 281: “Angiotensina” has been replaced with “Angiotensin”.
